# Progesterone receptor expression contributes to gemcitabine resistance at higher ECM stiffness in breast cancer cell lines

Emma Grant[1,2], Fatma A. Bucklain[1], Lucy Ginn[1¤], Peter Laity[3], Barbara Ciani[3], Helen E. Bryant🆔[1] *

**1** Department of Oncology and Metabolism, Academic Unit of Molecular Oncology, Sheffield Institute for Nucleic Acids (SInFoNiA), University of Sheffield, Sheffield, United Kingdom, **2** Department of Chemistry, Centre for Membrane Interactions and Dynamics (CMIAD), Krebs Institute, University of Sheffield, Sheffield, United Kingdom, **3** Department of Materials Science and Engineering, Natural Materials Group, Sheffield, United Kingdom

¤ Current address: Cell Signalling Group, Cancer Research UK Manchester Institute, University of Manchester, Alderley Park, Manchester, United Kingdom
* h.bryant@sheffield.ac.uk

**Data Availability Statement:** All relevant data are within the manuscript and its Supporting Information files.

## Abstract

Chemoresistance poses a great barrier to breast cancer treatment and is thought to correlate with increased matrix stiffness. We developed two-dimensional (2D) polyacrylamide (PAA) and three-dimensional (3D) alginate *in vitro* models of tissue stiffness that mimic the stiffness of normal breast and breast cancer. We then used these to compare cell viability in response to chemotherapeutic treatment. In both 2D and 3D we observed that breast cancer cell growth and size was increased at a higher stiffness corresponding to tumours compared to normal tissue. When chemotherapeutic response was measured, a specific differential response in cell viability was observed for gemcitabine in 2 of the 7 breast cancer cell lines investigated. MCF7 and T-47D cell lines showed gemcitabine resistance at 4 kPa compared to 500 Pa. These cell lines share a common phenotype of progesterone receptor (*PGR*) expression and, indeed, pre-treatment with the selective progesterone receptor modulator (SPRM) mifepristone abolished resistance to gemcitabine at high stiffness. Our data reveals that combined treatment with SPRMs may therefore help in reducing resistance to gemcitabine in stiffer breast tumours which are PGR positive.

## Introduction

In women, breast cancer (BC) is the leading cause of cancer deaths responsible for 15% of all 4.2 million female cancer deaths in 2018 [1]. It is the most commonly diagnosed cancer in women, accounting for 24.2% of all newly diagnosed female cancers [1]. The use of oestrogen receptor alpha (ERα) and epidermal growth factor receptor-2 (ERBB2/HER2) targeting agents has vastly improved outcomes but for some patients and often in advanced disease chemotherapeutics such as anthracyclines and taxanes are commonly used as standard therapy. However, primary and acquired resistance to both cytotoxics occurs limiting their success. New cytotoxic treatments are now available for patients who have been previously treated with anthracyclines

**Funding:** HB, BC and EG were funded by The University of Sheffield (https://www.sheffield.ac.uk/) and Team Verrico (https://www.teamverrico.org/). Microscopy was funded by The Wellcome Trust (https://wellcome.org/) grant GR077544AIA. PL was funded by European Union's Horizon 2020 (https://ec.europa.eu/programmes/horizon2020/en/home) research and innovation programme under grant agreement no. 713475. FB was funded by a King Abdul-Aziz University Scholarship. The funders had no role in study design, data collection and analysis, decision to publish, or preparation of the manuscript.

**Competing interests:** The authors have declared that no competing interests exist.

**Abbreviations:** 2D, two-dimensional; PAA, polyacrylamide; 3D, three-dimensional; *PGR*, progesterone receptor gene; PR, progesterone receptor protein; SPRM, selective progesterone receptor modulator; BC, breast cancer; ERα, oestrogen receptor alpha; ERBB2/HER2, epidermal growth factor receptor-2; ECM, extracellular matrix; EMT, epithelial to mesenchymal transition; YAP, yes-associated-protein; E-cadherin, epithelial-cadherin; Wnt, Wingless-related integration site; MAPK, mitogen-activated protein kinase; Bcl-xL, B-cell lymphoma-extra large; STAT3, signal transducer and activators of transcription 3.

and taxanes including gemcitabine. Gemcitabine (Gemzar; 2′, 2′-difluorodeoxycytidine) is an analogue of deoxycytidine and a pyrimidine antimetabolite widely used in other solid tumours. In clinical trials in BC patients, gemcitabine has produced varied results perhaps linked to BC subtype [2, 3]. In pancreatic cancer, where gemcitabine is a first line therapy in advanced disease, response is linked to tissue stiffness [4], but this has not been tested in BC. Given the varied response of BC to gemcitabine and to understand further the potential utility of gemcitabine in advanced BC we asked whether ECM stiffness modulates response to gemcitabine in different breast cancer cell lines.

Maintaining a homeostatic environment within a tissue requires dynamic conversation between epithelial cells and the cells which reside within the surrounding interstitial matrix. This includes fibroblasts which excrete ECM components allowing ECM modulation. This fine tuning of ECM content and structure provides an important balance of tension within the tissue. In glandular tissues, such as the breast, the tensional homeostasis is compliant where the breast has an elastic modulus somewhere in the region of 100–200 Pascals (Pa) [5]. In cancer however, stiffness is often heightened, where breast stiffness is increased from 100–200 Pa to ~4 kPa [5]. Increased extracellular matrix (ECM) stiffness is associated with increased resistance to chemotherapy. Indeed in breast cancer, patients with a lower breast elastography responded better to neoadjuvant chemotherapy than those with a higher measured elastography [6–9], suggesting that stiffness may be linked to chemotherapeutic response. We hypothesis that targeting ECM stiffness or the molecular pathways altered in response to stiffness, may be provide a novel therapeutic window for specific treatment of cancer vs normal cells. Alternatively targeting stiffness induced pathways could modulate response to cytotoxic agents.

In pancreatic cancer, stiffer tissues show resistance to gemcitabine [4], here we show a similar finding in BC. In addition this resistance appears dependent on PR signalling, suggesting that combined treatment with SPRMs may improve response to gemcitabine.

## Materials and methods

### Materials

**Cytotoxic agents and drugs.** Gemcitabine, 5-fluorouracil, hydroxyurea and mifepristone were purchased from Sigma Aldrich (UK). Phalloidin-488 was purchased from Santa Cruz Biotechnologies (TX, USA).

**Antibodies.** Anti-vinculin mouse antibody (MAB68961, R&D Systems, Min, USA), anti-E-cadherin rabbit antibody (24E10, Cell Signaling Technology, UK), anti-β-catenin mouse antibody (sc-7963, Santa Cruz Biotechnology, TX, USA), anti-yes kinase-associated protein (YAP) rabbit antibody (sc-154-07, Santa Cruz Biotechnology, TX, USA), anti-RPA34 mouse antibody (NA19L, Merck Life Science, UK) and anti-phosphorylated H2A histone family member X (γH2AX) S139 rabbit antibody (2577, Cell Signaling Technology, UK) were used for immunofluorescence and immunoblot analysis.

### Cell culture

Cell lines were obtained from DSMZ Cell Culture Collection or ATCC. CAL-51 (ACC 302), MCF-7 (ACC 115) and ZR-75-1 (CRL-1500) cell lines were cultured in high glucose Dulbecco's Modified Eagle Medium (DMEM) containing L-glutamine (Sigma Aldrich, UK). The MDA-MB-468 (HTB-132) and T-47D (HTB-133) cell lines were cultured in Roswell Park Memorial Institute (RPMI)-1640 medium containing L-glutamine (Sigma Aldrich, UK). The SK-BR-3 (HTB-30) cell line was cultured in high glucose McCoy's 5A (modified) medium containing L-glutamine (Lonza, CH). All media was supplemented with 10% foetal calf serum and 1x non-essential amino acids (Sigma Aldrich, UK).

## Models and rheology

**Polyacrylamide hydrogels.** Bottom coverslips were activated prior to pouring of poly-acrylamide gels with the addition of 70% industrial methylated spirits (IMS), 0.1 M sodium hydroxide, 0.5% v/v 3-aminopropyl-trimethoxysilane and 0.5% v/v glutaraldehyde. Top cover-slips were treated with 70% IMS and a thin layer of Rain-X (ITW Global Brands, TX, USA). Polyacrylamide gels comprising of varying acrylamide and bis-acrylamide percentages were set between the bottom and top coverslips on their reactive sides. After 30 minutes gelation, gels were soaked in phosphate buffered saline (PBS) for 20 minutes. Top coverslips were removed and collagen I was conjugated to the top surface of the gels using Sulphosuccinimidyl 6-(4′-azido-2′-nitrophenylamino)hexanoate (Sulfo-SANPAH) (ThermoFisher Scientific, UK). Gels were sterilised with 30 minutes of UV radiation and stored in PBS for up to 3 weeks prior to use. Prior to cell seeding, gels were blocked with 1% ethanolamine in HEPES buffer for 30 minutes at 4°C followed by serum free medium for at least 1 hr at 37°C.

**Alginate hydrogels.** A stock solution of 3% alginate was made by dissolving alginic acid sodium salt in 0.15 M sodium chloride solution. The solution was sterilised by autoclaving and then diluted with complete DMEM to form stock solutions of 1% and 2% alginate respectively. Alginate solutions were warmed to 37°C in the incubator for 15–30 minutes prior to use. Cells were mixed with the alginate by gentle pipetting at a concentration of 500,000 cells/mL of alginate for the MCF-7 cell line. Alginate-cell solutions were then pipetted dropwise into 200 mM calcium chloride solution in a 24-well plate using a multichannel pipette and low retention pipette tips. Alginate beads were incubated at 37°C for 10 minutes and then washed four times in complete DMEM. Gels were incubated in complete DMEM at 37°C prior to downstream assays.

**Rheology.** Rheology of hydrogels was measured on a Bohlin Gemini 200 rheometer fitted with a 25 mm diameter flat plate for polyacrylamide gels and a 10 mm diameter flat plate for alginate gels. The Peltier heating stage was set at a temperature of 37°C and axial closing forces of 0.2–5 N were applied. A frequency sweep from 10 to 0.01 Hz was implemented for a total of 11 frequency steps at an oscillatory strain of 0.02%. The hydrogel was flooded with distilled water and covered with an environmental cuff to prevent the sample from drying.

## Cell viability assay

Cells were plated on polyacrylamide (PAA) hydrogels or set into alginate beads and then incubated in complete culture medium at 37°C for 24 hrs. Medium was then removed and fresh medium added followed by addition of the cytotoxic agent or vehicle control at the appropriate concentration. The cells were incubated at 37°C for 48 hrs with cytotoxic agents before the medium was removed and replaced with fresh medium containing the cytotoxic agent or vehicle control. Cells were then incubated at 37°C for 24 hrs prior to the measurement of cell viability using alamarBlue. To undertake an alamarBlue reading cell medium was removed from the cells and replaced with fresh medium containing alamarBlue reagent diluted 1:10. The cells were incubated for 3–4 hrs at 37°C protected from light. After incubation, 100 μL of the solution was moved to a 96-well plate and then read on a SpectraMax M5e multi-mode microplate reader at an excitation of 570 nm and an emission of 600 nm using SoftMax Pro software.

## Immunofluorescence

Cells were stained with antibodies against vinculin, E-cadherin, β-catenin, YAP, RPA34 and γH2AX using immunofluorescence. Briefly, cells were fixed with 4% paraformaldehyde in PBS for 10 minutes at room temperature. Cells stained for RPA34 were first pre-extracted with ice-cold extraction buffer (100 mM NaCl, 300 mM sucrose, 3 mM magnesium chloride, 10 mM piperazine-N,N′-bis(2-ethanesulphonic acid) (PIPES) pH 6.8, 0.5% Triton-X-100) for 2

minutes at 4˚C. Cells were then blocked and permeabilised with 0.1–0.5% v/v Triton-X-100 with 1–3% BSA in PBS/TBS for 10 minutes and then incubated with primary antibody diluted in 1% w/v BSA in PBS/TBS for 16 hrs at 4˚C in a humidified chamber. Cells were washed 3 times prior to incubation with Alexa Fluor®–conjugated secondary antibody diluted in 1% w/v BSA in PBS/TBS with 1 μg/mL DAPI for 1 hr at room temperature protected from light. Coverslips were washed 3 times and then mounted onto glass slides with a solution of 90% glycerol. Slides were stored at 4˚C protected from light prior to analysis. Actin and Vinculin images were acquired at the Wolfson Light Microscopy Facility using the Applied Precision deconvolution DeltaVision microscope (GE Healthcare Life Sciences, UK). A UPlanSAPO 40x oil objective lens was employed to image z-stack images using SoftWorks software. All other images were acquired on a Nikon TE200 inverted fluorescent microscope using Volocity Software. All images were analysed using FIJI software (**https://imagej.net/ImageJ**).

## Western blotting

MCF-7 cells cultured on PAA hydrogels were inverted onto 1 x RIPA lysis buffer (700 μL ddH2O, 200 μL 5x RIPA lysis buffer (250 mM tris pH 8.0, 750 Mm NaCl, 0.5% SDS, 5% NP-40 alternative, 2.5% sodium deoxycholate), 10 μL 100 mM PMSF, 10 μL 100x protease inhibitor, 10 μL 100x phosphatase inhibitor and 2 μL Benzonase per 1 mL) for 1 minute at 4˚C. The solution incubated with gels was removed, incubated on ice during which it was vortexed every 10 minutes for 30 minutes and then passed through a 25G needle 10 times. This solution was then centrifuged for 10 minutes at 13,400 RPM at 4˚C. The supernatant was removed and lysates were separated by sodium dodecyl sulfate-polyacrylamide gel electrophoresis (SDS-PAGE). Proteins were then transferred onto 0.45 μm Protran nitrocellulose transfer membrane (GE Healthcare) on ice at 85 V for 2 hrs. Membranes were blocked with 5% w/v milk in TBS for 1 hr at room temperature. Membranes were then probed with primary antibody diluted in 5% w/v milk in TBS at 4˚C for 16–24 hrs. Membranes were washed three times with 0.05% v/v Tween-20 in TBS (TBS-T) every 10 minutes. Membranes were then incubated with the appropriate HRP-labelled secondary antibody diluted in 5% w/v milk in TBS for 1 hr at room temperature. Following secondary antibody incubation membranes were washed three times with TBS-T every 10 minutes prior to chemiluminescent detection.

## Real time PCR

RNA was extracted from MCF-7, T-47D, ZR-75-1, CAL-51, MDA-MB-231 and MDA-MB-468 cell lines using a Qiagen RNeasy Plus Mini kit following the manufacturers protocol and stored at -80˚C. Three samples were collected for each cell line. RNA samples were reverse transcribed into cDNA by use of a Qiagen RT$^2$ first strand kit following the manufactures protocol. cDNA was quantified using Qiagen RT2 SYBR® Green qPCR master mixes following the manufacturers protocol. Primers were used against total PGR (PPH01007F), ER (PPH01001A) and HER2 (PPH00209B) probes and a β-actin control (PPH00073G). Alternatively RNA was extracted from cells grown on polyacrylamide gels using using Trizol$^{TM}$ reagent and Phasemaker tubes (Thermo Fisher Scientific, UK). RNA was them precipitated with isopropanol/ethanol before resuspending in RNAse free water. This was then reverse transcribed into cDNA using SuperScript IV Reverse Transcriptase (Thermo Fisher Scientific, UK). For total PGR (Hs01556702_m1), PGR B (Hs04419616_s1) or a GAPDH control (Hs02786624_g1) TaqMan probes (Thermo Fisher Scientific, UK) were used with TaqMan™ Gene Expression Master Mix (Thermo Fisher Scientific, UK). *PGR-A* specific probes cannot be designed due to complete overlap with *PGR-B* so *PGR-A* expression was determined by subtracting the fold expression change of *PGR-B* from that of total *PGR* expression.

## Statistics

Results were determined to be normally distributed or not using Shapiro-Wilk test for normality, prior to analysis with a paired Student's t-test or a Mann-Whitney U or Kruskal-Wallis test as relevant and indicated. P values below 0.05 were considered representative of data that were significantly different. For parametric (normal) data the mean ± the standard deviation (SD) or the standard error of the mean (SEM) are presented. For non-parametric data the median is plotted. Microsoft Excel and Graphpad Prism 8 software were used for analysis of all data.

## Results

### MCF-7 breast cancer cells increase in surface area and cell growth during the first 72 hrs at a higher stiffness

The response of breast cancer cell lines to matrix stiffness was first investigated in a 2D polyacrylamide (PAA) model, coated with collagen I (Fig 1A). Two PAA gels were used, the first representing the stiffness of normal breast tissue (500 Pa), and the second representing cancerous breast tissue (4 kPa) [5, 6] (S1 Fig). When cultured on the PAA gels, MCF-7 breast cancer cells showed different morphology between the stiffnesses (Fig 1B). Immunofluorescent staining of the actin cytoskeleton and DNA (Fig 1C) determined that cells had an increase in cytoplasmic and nuclear area at 4 kPa, compared to 500 Pa (Fig 1D), consistent with the literature [7]. Staining of mature focal adhesions (vinculin) also showed a change in the staining pattern (Fig 1C). Cells at 500 Pa had a small number of vinculin foci at the cell periphery, while at 4 kPa vinculin staining appears to be a pan stain, indicating a difference in adhesion between the two stiffnesses. Cell viability assays and cell counting determined that cell growth increased in cells grown at 4 kPa compared to 500 Pa for the first 72 hrs of growth on the matrix (Fig 1E and 1F). Growth rate became parallel between 72–144 hrs, perhaps suggesting an adaptation to the matrix, or a difference in lag time at low stiffness. An increased growth rate at higher stiffnesses has been observed previously in other stiffness models [8, 9], further validating our model.

### MCF-7 breast cancer cells are resistant to gemcitabine at an increased stiffness but this in not attributed to changes in EMT markers

Cell viability assays were used to determine the response of MCF-7 cells to three anti-metabolites, gemcitabine, 5-FU and HU at both 500 Pa and 4 kPa (Fig 2A). A differential response was only observed to gemcitabine, where cells cultured at 4 kPa were more resistant to gemcitabine than cells cultured at 500 Pa. Interestingly this was not observed for the other antimetabolite agents tested, 5-FU and HU (Fig 2A). In other models, stiffness has been seen to promote elements of EMT, including decreases in E-cadherin expression, nuclear localisation of β-catenin, YAP and TAZ and changes in cell shape towards a mesenchymal phenotype [10, 11]. Further these changes have been linked to therapeutic response [10, 11]. However, although we did see changes in cell morphology (Fig 1), we observed no difference in β-catenin localisation or E-cadherin expression between the two stiffnesses (Fig 2B and 2C). Similarly, there was no difference in the nuclear intensity of YAP between the two stiffnesses (Fig 2E and 2F). These results suggest that such EMT-like characteristics are not being altered in MCF-7 cells between 500 Pa and 4 kPa.

### Resistance to gemcitabine at 4 kPa is not associated with changes in replication stress or DNA damage

As gemcitabine is an antimetabolite drug which functions by increasing replication stress, we next determined how replication stress and DNA damage were altered between the two

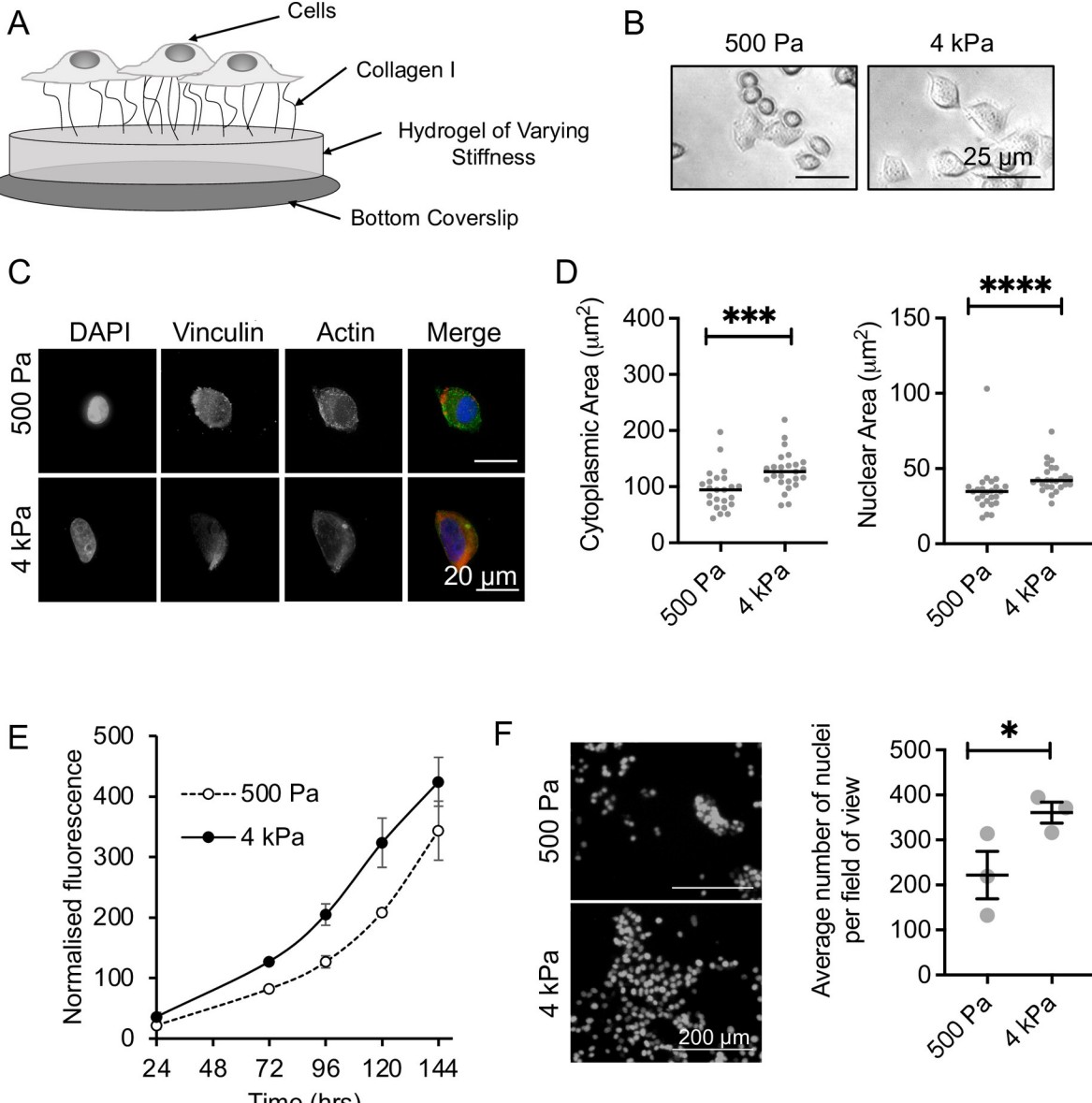

**Fig 1. Validating the polyacrylamide (PAA) hydrogel model.** (A) Schematic of the PAA model. (B) Brightfield microscopy of MCF-7 cells cultured for 48 hrs on the PAA model. (C) Immunofluorescent imaging of vinculin and actin in MCF-7 cells cultured on 500 Pa and 4 kPa stiffness hydrogels for 16 hrs. (D) Median and individual values for nuclear and cytoplasmic areas counting for 17–25 cells per condition. Significance was calculated using a Mann-Whitney U-test. (E) MCF-7 cell viability as measured by alamarBlue assay over 144 hrs culture on 500 Pa and 4 kPa stiffness hydrogels. The mean and SEM of 2 independent repeats are shown. (F) MCF-7 cell growth after 120 hrs culture on 500 Pa and 4 kPa stiffness hydrogels as measured by DAPI stained nuclei. The mean and SEM of 3 independent repeats are shown. Significance was calculated using a Student's paired t-test.

stiffnesses. The formation of RPA34 foci was used as a marker of replication stress and investigated by immunofluorescent staining (Fig 3A). The addition of gemcitabine resulted in a large increase in RPA34 intensity at both stiffnesses as expected and confirmed gemcitabine function in our model. However, there was no difference in RPA34 intensity between the two stiffnesses post gemcitabine treatment, suggesting that stiffness does not alter the level of gemcitabine induced replication stress. DNA damage was determined through immunofluorescent staining for γH2AX Ser139 (Fig 3B). There was a significant increase in the intensity of

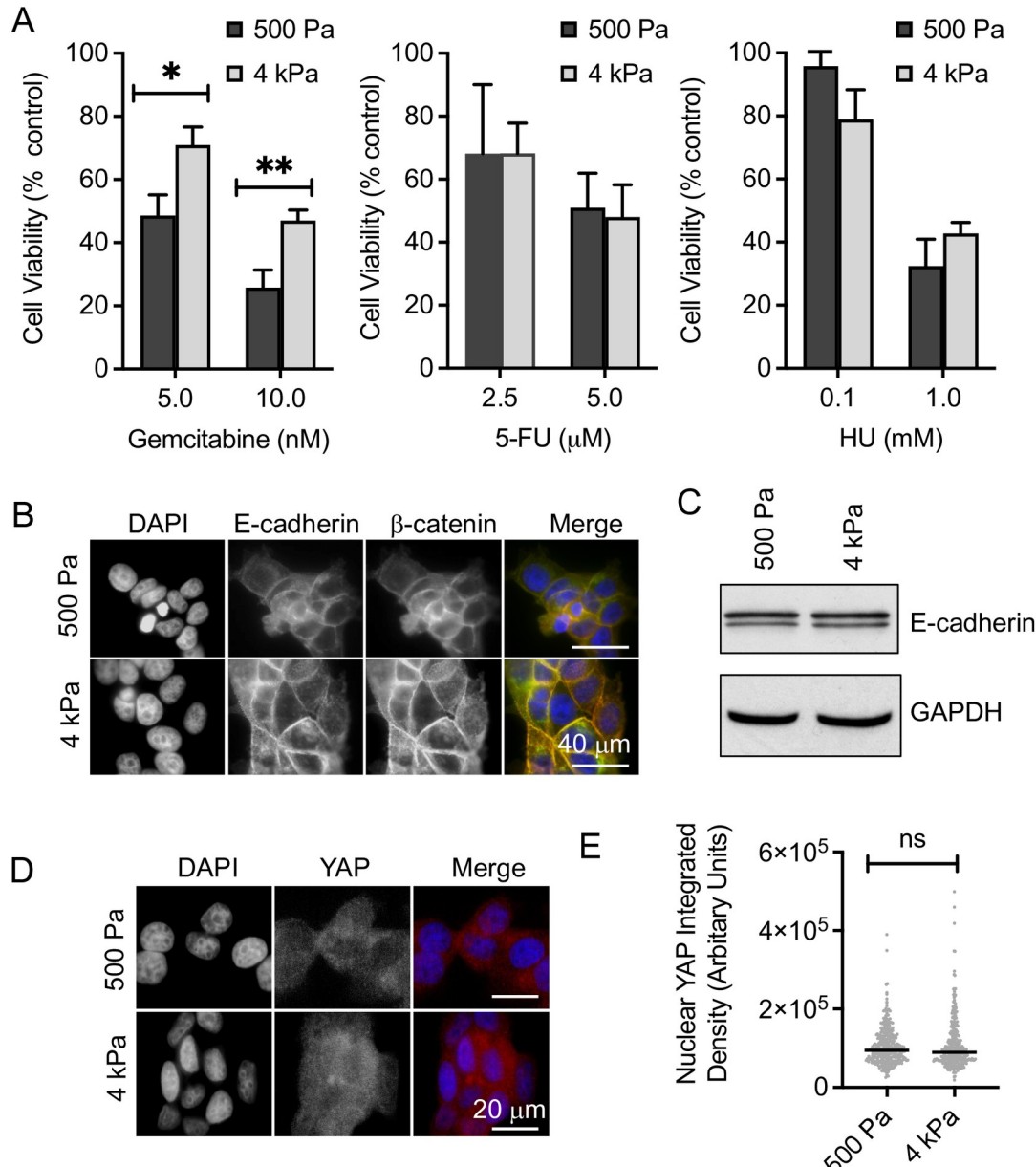

**Fig 2. Matrix stiffness modulates chemotherapeutic resistance to gemcitabine but this is not attributed to classical EMT.**
(A) MCF-7 cells were cultured on 500 Pa and 4 kPa hydrogels for 24 hrs prior to the addition of chemotherapeutic agents as indicated; 5-fluroruracil (5-FU), hydroxyurea (HU), and gemcitabine. After 72 hrs cell viability was measured by alamarBlue assay. The means and SEMs are plotted for >3 independent repeats. Significance was calculated using a Student's paired t-test. (B) E-cadherin and β-catenin visualised by immunofluorescence in MCF-7 cells cultured on 500 Pa and 4 kPa stiffness hydrogels for 24 hrs. (C) Western blot for E-cadherin and a GAPDH control of whole cell extracts from MCF-7 cells cultured on 500 Pa and 4 kPa stiffness hydrogels for 72 hrs. (D) Immunofluorescent visualisation of YAP in MCF-7 cells cultured on 500 Pa and 4 kPa stiffness hydrogels for 24 hrs. (E) Nuclear YAP intensity (integrated density) measured in approximately 100 nuclei for each condition for each of 3 independent repeats. The median and individual cells from pooled data are plotted. Significance was calculated using a Mann-Whitney U-test.

basal levels of γH2AX in cells at 4 kPa compared to 500 Pa, suggesting that stiffness contributes to an increase in endogenous DNA damage. The addition of gemcitabine also led to a significant increase in the intensity of γH2AX for cells cultured on both stiffnesses, further confirming gemcitabine function. However, there was no difference in γH2AX intensity between cells

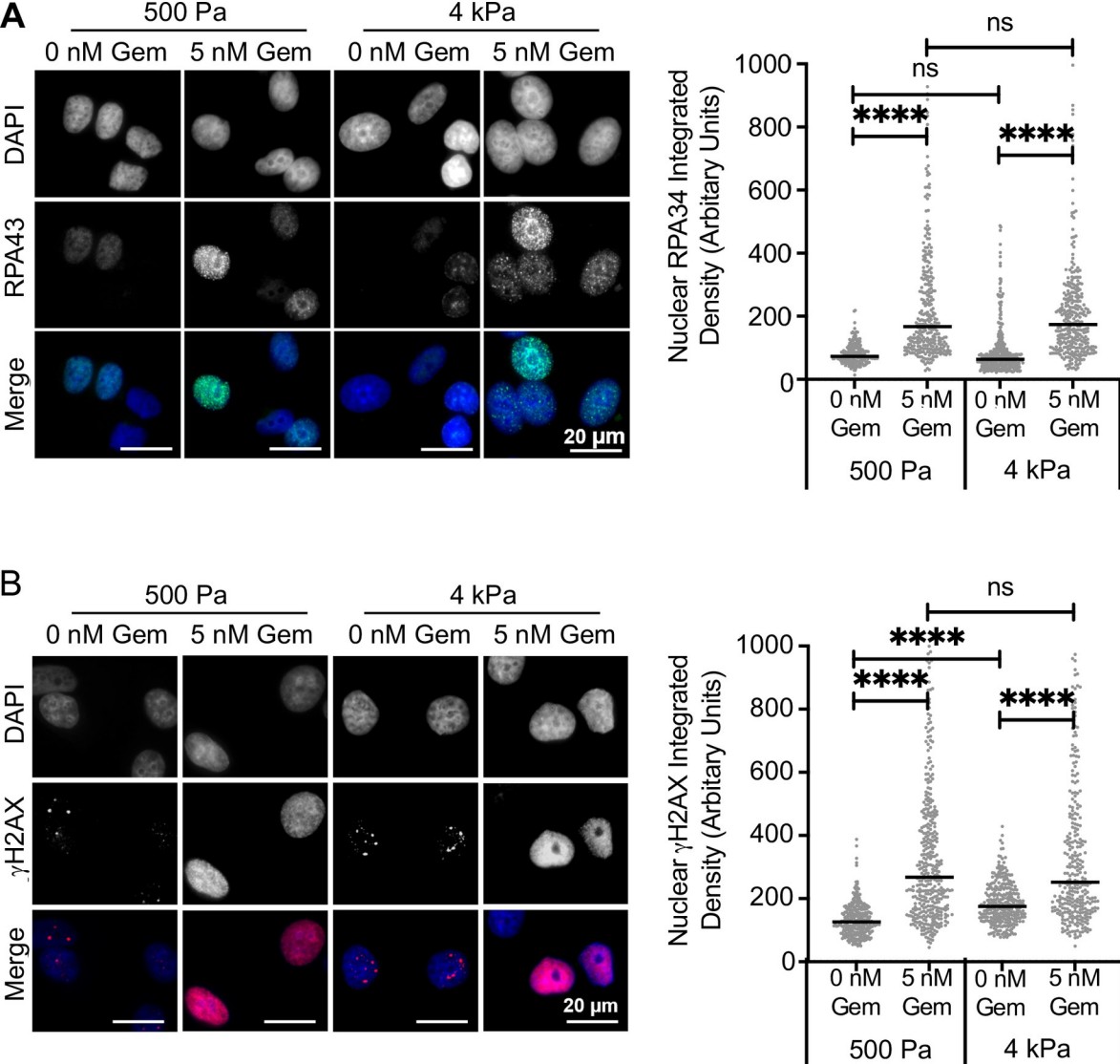

**Fig 3. Stiffness specific gemcitabine resistance is not associated with altered levels of replication stress or DNA damage levels.** The MCF-7 cell line was cultured on 500 Pa and 4 kPa stiffness hydrogels for 24 hrs prior to treatment with 5 nM gemcitabine (Gem) or a vehicle control for a further 24 hrs. Cells were then fixed and stained using immunofluorescence for (A) RPA34 or (B) γH2AX. Representative images are shown for each stain. Nuclear intensity (integrated density) was measured in 100 nuclei for each of 3 independent repeats. The median and pooled data from the repeats are shown. Statistical significance was calculated using a Kruskal-Wallis test.

cultured at 500 Pa and 4 kPa following gemcitabine treatment, suggesting that stiffness does not alter gemcitabine induced DNA damage.

## Resistance to gemcitabine is observed only in progesterone receptor positive breast cancer cell lines

We next explored gemcitabine response in a panel of breast cancer cell lines with different p53 and hormone receptor statuses (Fig 4A–4E). The cell lines exhibited a range of sensitivities to gemcitabine, but resistance at higher stiffness, as seen in MCF-7 (Fig 2), was only observed in

the T47D cell line (Fig 4A). MCF-7 and T47D are the only cell lines tested that are *PGR* positive (Fig 4F), suggesting a correlation between *PGR* expression and gemcitabine resistance at higher stiffnesses.

The connection between *PGR* expression and gemcitabine resistance at 4 kPa was confirmed through use of the SPRM—mifepristone (Fig 5), where pre-treatment with 10–15 μM mifepristone was sufficient to abolish the stiffness specific resistance at 4 kPa.

## Resistance to gemcitabine at higher stiffnesses is also observed in a 3D alginate stiffness model

We used a 3D alginate bead model (Fig 6A) to further investigate the connection between substrate stiffness, *PGR* expression and gemcitabine resistance. MCF-7 cells were encapsulated in the beads where they grew into cell clusters over 96 hrs (Fig 6B). Gel stiffness was lower than the 2D PAA model, at 300 Pa and 1.9 kPa for 1% and 2% alginate respectively (Fig 6C and S2 Fig), but was still close to the measured normal and cancerous breast tissue [5, 6], and significantly different to each other allowing for a direct comparison. When the response to gemcitabine treatment was investigated in MCF-7 cells grown in alginate, resistance was observed at higher compared to lower stiffness, consistent with the PAA model (Fig 6D). In addition, stiffness dependent-resistance to gemcitabine was abolished following pre-treatment with mifepristone (Fig 6E). Notably, the relationship between substrate stiffness and gemcitabine and its reversal with mifepristone was only statistically significant at higher doses of gemcitabine and mifepristone. We speculate this is because of limited diffusion of the drugs in 3D.

## Discussion

In our 2D and 3D models, the SPRM mifepristone abolished the stiffness dependent resistance to gemcitabine seen in *PGR* expressing breast cancer cells. To our knowledge this is the first time a link between PRs, matrix stiffness and gemcitabine resistance has been determined. Combined treatment with SPRMs may therefore help in reducing resistance to gemcitabine in stiffer breast tumours that are *PGR* positive.

Progesterone signalling has been linked to stiffness previously, where increased tension during pregnancy results in an increase in production of basal ECM components and reduces levels of progesterone [12]. Conversely, the addition of progesterone reduced the production of other ECM components (fibronectin and laminin) [12], suggesting that PR signalling modulates ECM composition. Secondly, YAP acts as a co-activator of the PR [13], further linking stiffness to PR signalling. However, in our 2D model significant nuclear localisation of YAP was not observed suggesting that other mechanisms may be responsible.

Breast cancer resistance to doxorubicin at higher stiffnesses has previously been attributed to EMT [11]. We could not detect any change in the classical EMT markers, E-cadherin or YAP, between the two stiffnesses. However, classical EMT markers have recently been shown not to be accurate markers of EMT in breast cancer stiffness models of mechanotransduction [14], suggesting that EMT effects cannot be ruled out in our models.

Following PR inhibition, we observed that gemcitabine resistance was abolished at higher stiffnesses which may indicate that PR signalling is increased at higher stiffnesses. PR signalling modulates expression of various target proteins such as cyclin D1 and Wnt [15] and also acts within the cytoplasm together with other proteins such as c-Src to activate proliferative pathways such as MAPK [16]. Linked to this, gemcitabine treatment also activates MAPK in pancreatic cancer cell lines [17] and Wnt signalling has been linked to gemcitabine resistance in pancreatic cancer [18]. It has been reported that the PR isoforms, PR-A and PR-B, can also alter the expression of a number of different genes [19]. In particular, PR-A specifically induces

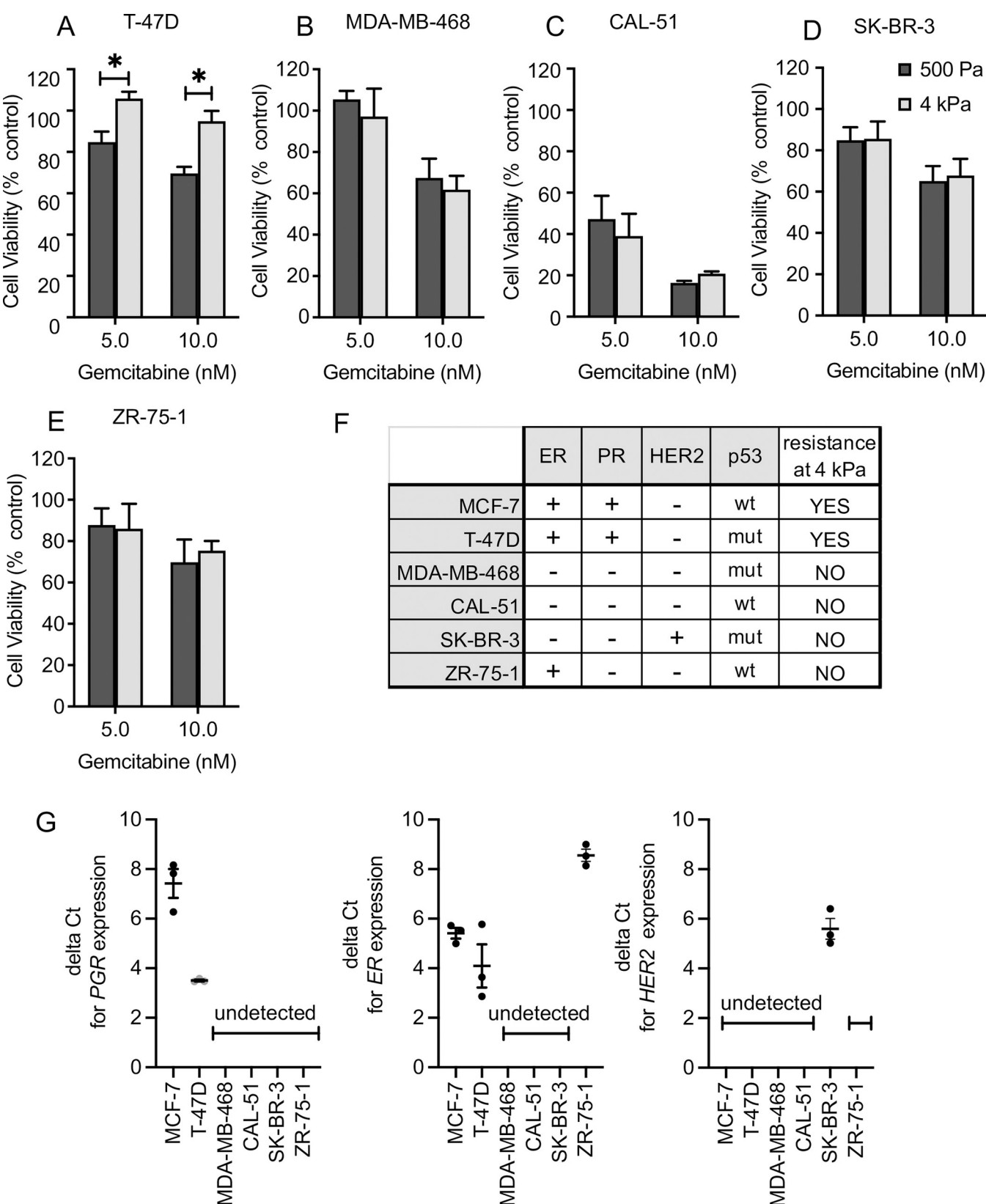

**Fig 4. Resistance to gemcitabine at higher stiffness occurred only in progesterone receptor positive cell lines.** (A-E) The T-47D, MDA-MB-468, CAL-51, SKBR-3, and ZR-75-1 cell lines were cultured on 500 Pa and 4 kPa stiffness hydrogels for 24 hrs prior to the addition of gemcitabine (gem). After 72 hrs

treatment cell viability was measured by alamarBlue assay. Means and standard errors of the mean are plotted for three independent repeats. Significance was calculated using a Student's paired T-test. (F) Summary of hormone receptor and p53 expression status in the breast cancer cell lines. (G) Hormone expression investigated relative to β-actin as determined by RT-q-PCR. The delta CT values for cells grown on glass, are plotted in grey for 3 independent cell pellets performed in triplicate. The mean and standard error of the mean are plotted in black.

overexpression of the bcl-2 family protein, Bcl-xL. Changes in PR isoform expression due to stiffnesses may therefore modulate gemcitabine resistance via Bcl-xL. Bcl-xL can be upregulated by STAT3 [20], a protein also upregulated by PR signalling [21] and attributed to gemcitabine resistance [22]. Furthermore, STAT3 and p300, which both act as PR co-activators [23, 24], have recently been identified as mechanotransducers in breast cancer cell lines [14], indicating a further link between *PGR* and stiffness in breast cancer. Here we examined expression of PGR and it's isoforms on different stiffnesses (S3 Fig), we saw reduction of total *PGR* and *PGR-B* in cells grown on stiffer matrixes, altering the ratio of PGR-A to PGR-B, again suggesting that the PR pathway is important in the context of stiffness. Extended studies of PR signalling in the context of stiffness will be the subject of future studies. In contrast ER expression was not altered (S3 Fig).

It should be noted that high dose 10–15 μM of mifepristone was used to see this effect. It is possible that at high concentration, mifespristone could have nonspecific effects through binding to other receptors such as glucocorticoid receptor (GR). It will be important going forward

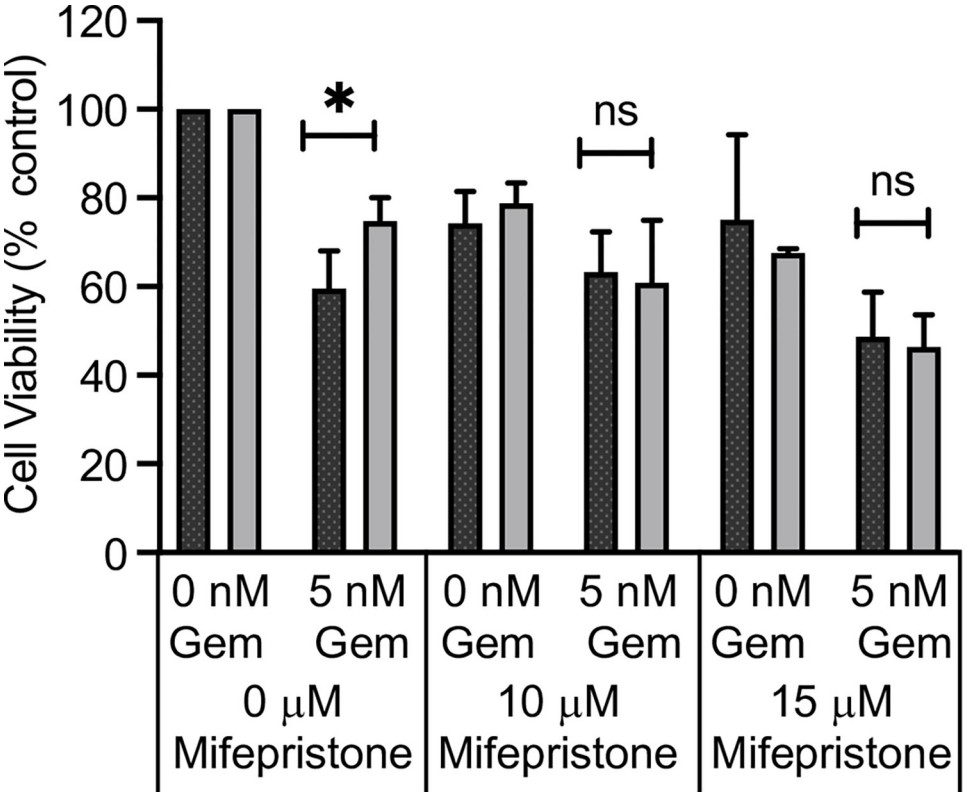

**Fig 5. Resistance to gemcitabine can be reversed by addition of the SPRM–mifepristone.** The MCF-7 cell line was cultured on 500 Pa and 4 kPa hydrogels for 24 hrs prior to the addition of mifepristone as indicated for 24 hrs. Gemcitabine was then added in combination with mifepristone or its vehicle control for a further 72 hrs. Cell viability was measured by alamarBlue assay. The means and SEMs are plotted for 3 independent repeats. Significance was calculated using a Student's paired t-test.

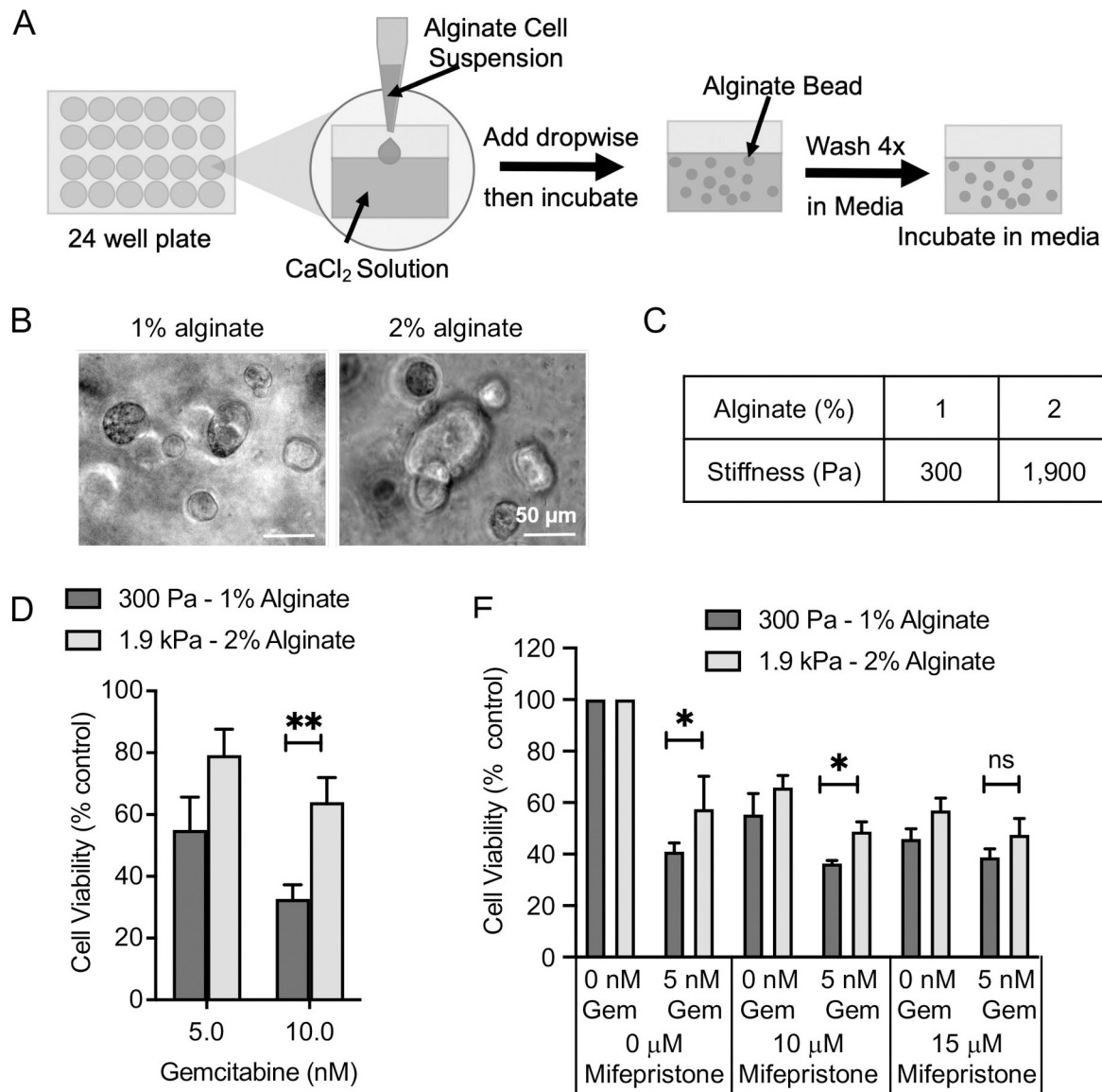

**Fig 6. Resistance to gemcitabine at higher stiffnesses also occurs in a 3D model and is reversed by mifepristone.** (A) Schematic of the alginate model where different alginate percentages are used to give different stiffnesses. Beads were formed by dropping alginate cell suspension into calcium chloride solution. Alginate beads were then washed in medium and cells allowed to grow by incubating at 37˚C. (B) Representative brightfield images of MCF-7 cells 96 hrs post alginate bead gelation. (C) Rounded average measured elastic modulus measured for 6 gels pooled from 2 independent repeats measured 1 hr after gelation at a frequency of 1 Hz by rheometry. (D) MCF-7 cells were cultured in alginate beads for 24 hrs prior to the addition of gemcitabine, after 72 hrs treatment cell viability was measured by alamarBlue assay. The means and SEMs are plotted for 4 independent repeats. (E) MCF-7 cells were cultured in alginate hydrogels for 24 hrs prior to the addition of mifepristone. After 24 hrs 5 nM gemcitabine (Gem) was added. Cells were incubated for 72 hrs before cell viability was measured by alamarBlue assay. The means and SEMs are plotted for 3 independent repeats. Significance was calculated using a Student's paired t-test.

to examine other pathways by which mifepristone could be functioning. It would also be interesting to test whether protein agonists such as medroxyprogesterone or R5020 reverse mifepristone's effect.

The cell lines where resistance is observed are both ER and PR expressing. There are known links between ER and PR function especially with regards to endocrine therapy resistance [25].

Although resistance was not seen in cell lines only expressing ER, ER expression in PR/ER expressing cells was not altered on stiffer matrices and the SPRM mifepristone abolished resistance. The study is limited by lack of inclusion of a PR only expressing cell line and we cannot exclude the possibility that cross-talk between PR and ER activity contributes to the phenotype seen.

Gemcitabine is not used as a first line standard of care treatment in the ER/PR positive tumours that our cell lines represent. However, PR is present in 55–60% of all breast cancers [26] and clinical trials are currently underway for PR inhibition as a monotherapy [27] indicating an exciting new treatment regime. Our findings suggest that combination with gemcitabine would be an exciting translational possibility for patients bearing PR expressing tumours.

## Conclusion

Our findings show increased resistance to gemcitabine when PR positive cells are grown at higher stiffnesses such as those found in tumours. In addition, they indicate a novel connection between PR signalling and stiffness in chemotherapeutic resistance. Future investigation into PR isoform expression, PR activity and signalling will provide further insights into the molecular mechanisms underlying this connection, potentially elucidating novel therapeutic combinations for breast cancer patients with PR expressing tumours.

## Supporting information

**S1 Fig. Rheological measurements of polyacrylamide hydrogels.** (A) A representative example of rheological graphs for each gel composition showing elastic modulus, viscous modulus and phase angle. Rheological measurements were made on a Bohlin Gemini 200 rheometer fitted with a 25 mm diameter flat plate at 37°C. An axial closing force of 0.2–5 N was applied to each hydrogel and the mechanical response was measured over an oscillating frequency range of 10 to 0.01 Hz at a strain of 0.02%. (B) Average elastic modulus ± standard deviation for 3 gels measured over 3 independent repeats for the 5.42% and 7.46% acrylamide gels at a frequency of 1 Hz. (C) Average elastic modulus ± standard deviation for 3 gels measured over 3 independent repeats for the 5.42% and 7.46% acrylamide gels at a frequency of 0.4 Hz. (DOCX)

**S2 Fig. Rheological measurements of alginate beads.** (A) A representative example of rheological graphs for each alginate percentage at 1 hr post gelation showing elastic modulus, viscous modulus and phase angle. Rheological measurements were made on a Bohlin Gemini 200 rheometer fitted with a 10 mm diameter flat plate at 37°C. An axial closing force of 0.2–5 N was applied to each hydrogel and the mechanical response was measured over an oscillating frequency range of 10 to 0.01 Hz at a strain of 0.02%. All gels were made using 200 mM calcium chloride and 4 media washes. (B) Average measured elastic modulus ± standard deviation for 6 gels pooled from 2 independent repeats for the two gel compositions 1 hr after gelation at a frequency of 1 Hz. Each repeat is plotted in a different colour and the mean and standard deviation is plotted in black. (DOCX)

**S3 Fig. RT-PCR analysis of PR and ER expression in cells grown on 500 Pa and 4KPa stiffness gels.** Expression of the total progesterone receptor gene (*PGR*) and the separate A and B isoforms for MCF-7 cells by q-RT-PCR for one independent repeat. Cells were cultured on 500 Pa or 4 kPa stiffness hydrogels for 72 hrs prior to RNA extraction. Expression of *ER* total *PGR* and *PGR-B* were measured relative to a Glyceraldehyde 3-phosphate dehydrogenase (*GAPD*) internal control. Fold change in expression was calculated relative to 500 Pa. *PGR-A*

expression was determined by subtracting the fold expression change of *PGR-B* from that of total *PGR* expression.
(DOCX)

**S1 Raw images.**
(PDF)

## Acknowledgments

The authors wish to thank the following members of The University of Sheffield, Dr Darren Robinson, Wolfson Light Microscopy Facility for training and advice on imaging, Dr Chris Holland Natural Materials Group, Department of Materials Science and Engineering for access to the Rheology equipment and training and Dr Polly Gravells, Oncology and Metabolism for advice, discussions and proof-reading of the manuscript.

## Author Contributions

**Conceptualization:** Barbara Ciani, Helen E. Bryant.

**Data curation:** Emma Grant, Fatma A. Bucklain, Lucy Ginn.

**Formal analysis:** Emma Grant, Fatma A. Bucklain, Lucy Ginn.

**Funding acquisition:** Barbara Ciani, Helen E. Bryant.

**Investigation:** Emma Grant, Fatma A. Bucklain, Lucy Ginn.

**Methodology:** Emma Grant, Fatma A. Bucklain, Lucy Ginn, Peter Laity.

**Project administration:** Helen E. Bryant.

**Supervision:** Peter Laity, Barbara Ciani, Helen E. Bryant.

**Writing – original draft:** Helen E. Bryant.

**Writing – review & editing:** Emma Grant, Lucy Ginn, Peter Laity, Barbara Ciani, Helen E. Bryant.

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
