## [Decision Letter · Decision Letter 0]

16 Dec 2021

PONE-D-21-34392Progesterone receptor expression contributes to gemcitabine resistance at higher ECM stiffness in breast cancer cell lines.PLOS ONE

Dear Dr. Bryant,

Thank you for submitting your manuscript to PLOS ONE. After careful consideration, we feel that it has merit but does not fully meet PLOS ONE’s publication criteria as it currently stands. Therefore, we invite you to submit a revised version of the manuscript that addresses the points raised during the review process.

Both reviewers identified information that is missing in the manuscript including direct evidence for receptor status in cell lines used, and a general lack of mechanistic data to explain hormone resistance. One reviewer wanted to see a more comprehensive introduction section. Rationales for experimental design including choice of cell lines should be better explained, and insufficient data to support the criteria for EMT were provided. 

We look forward to receiving your revised manuscript.

Kind regards,

Philip C. Trackman, Ph.D.

Academic Editor

PLOS ONE

Journal Requirements:

Reviewers' comments:

Reviewer's Responses to Questions

**Comments to the Author**

1. Is the manuscript technically sound, and do the data support the conclusions?

Reviewer #1: Partly

Reviewer #2: Yes

2. Has the statistical analysis been performed appropriately and rigorously? 

Reviewer #1: Yes

Reviewer #2: Yes

3. Have the authors made all data underlying the findings in their manuscript fully available?

Reviewer #1: Yes

Reviewer #2: Yes

4. Is the manuscript presented in an intelligible fashion and written in standard English?

Reviewer #1: Yes

Reviewer #2: Yes

5. Review Comments to the Author

Reviewer #1: According to research ZR-75-1 typically can expresses both ER and PR. There are variants of ZR-75 which do not express PR, however, this variant should have PR expression. Did authors confirm loss or PR in their line through protein expression? Do the authors have an explanation for loss of PR in this line and was ER expression validated for this line?

PR expression is tightly linked through ER activity and prior studies demonstrate resistance to endocrine theories through stiff matrix adhesion and activation of pro survival pathways. Do the authors suggest that chemo resistance is PR mediated alone or is there relevance to ER in this dynamic? This aspect should be expanded on.

To fully demonstrate the link is not ER mediated in some way, Authors should show if PR expression changes on stiff matrix vs soft matrix for ER positive cell lines. Further ER expression and ERE activity should be confirmed for cell lines on stiff matrix.

Authors demonstrate chemotherapy resistance in model luminal A lines, the standard of care treatment is conventionally endocrine therapy for model ER expressing cell lines. The authors should expand on and highlight the significance of sensitizing cells to chemotherapy when first line may be an endocrine therapy and not a chemotherapy.

Reviewer #2: Manuscript Number: PONE-D-21-34392

Title: Progesterone receptor expression contributes to gemcitabine resistance at higher ECM

stiffness in breast cancer cell lines.

General comments

This manuscript examined the correlation between matrix stiffness and sensitivity to chemotherapy in breast cancer using two-dimensional (2D) and three-dimensional (3D) culture systems. The authors showed that breast cancer cell growth and size increased at a higher stiffness in 2D and 3D culture systems. Interestingly, the authors found that MCF-7 and T47D breast cancer cells expressing progesterone receptors, PR, showed gemcitabine resistance at higher stiffness (4 kPa) than lower stiffness (500 Pa). Pre-treatment with a progestin antagonist, mifepristone (RU486), abolished gemcitabine when cells were grown at high stiffness (kPa). The authors suggested that selective progesterone receptor modulators (SPRM) such as mifepristone may be helpful in reducing resistance to gemcitabine in PR-positive breast tumors.

Overall, the author described an exciting discovery that PR signaling could play a role in sensitivity to a chemotherapeutic agent such as gemcitabine. However, the data presented in this manuscript is rather descriptive with little to no mechanistic detail, and some key points will need to be clarified, especially regarding PR contribution. A major revision of the manuscript is suggested.

Specific comments

1. Introduction: additional information is needed. The background and significance of the study is not clear. What is known about matrix and cancer cells? What is the significance of this study?

2. How gemcitabine affects breast cancer is not known. It is not clear what the rationale for investigating the role of gemcitabine in breast cancer is.

3. Line 240. The conclusion “These results suggest that EMT is not occurring in MCF-7 cells between 500 Pa and 4 kPa.” needs to be further substantiated. The authors only examined two EMT markers in breast cancer cells and concluded that EMT was not occurring. More information is required.

Additionally, EMT usually occurs in nontransformed cells. What is the significance of

4. PR contribution to gemcitabine sensitivity. High dose 10-15 uM of mifepristone is needed to see the effect. Mifespristone is a PR antagonist that has a high affinity to PR and should bind to the receptor in nM range. At high concentration, mifespristone could have nonspecific effects through binding to other receptors such as glucocorticoid receptor (GR). Additionally, could progetin agonist such as medroxyprogesterone or R5020 reverse mifepristone’s effect.

6. PLOS authors have the option to publish the peer review history of their article (what does this mean?). If published, this will include your full peer review and any attached files.

Reviewer #1: No

Reviewer #2: No

---

## [Author Response · Author response to Decision Letter 0]

25 Mar 2022

Reviewer 1 says that the data are partially technically sound. In line with PlosOne criteria We have focused mainly on addressing this point with additional experiments. We have also included context and further discussion of the data and conclusions as requested.

Reviewer #1: According to research ZR-75-1 typically can expresses both ER and PR. There are variants of ZR-75 which do not express PR, however, this variant should have PR expression. Did authors confirm loss or PR in their line through protein expression? Do the authors have an explanation for loss of PR in this line and was ER expression validated for this line?

• PR expression was confirmed by RT-PCR but not by protein expression. We also tested ER and ERB2 (HER2) expression in the cells by RT-PCR these data are now included in revised figure 4. 

Reviewer #1: PR expression is tightly linked through ER activity and prior studies demonstrate resistance to endocrine theories through stiff matrix adhesion and activation of pro survival pathways. Do the authors suggest that chemo resistance is PR mediated alone or is there relevance to ER in this dynamic? This aspect should be expanded on. 

To fully demonstrate the link is not ER mediated in some way, Authors should show if PR expression changes on stiff matrix vs soft matrix for ER positive cell lines. Further ER expression and ERE activity should be confirmed for cell lines on stiff matrix.

• We agree there is a link between PR and ER in relation to endocrine therapy and have now discussed the possibility of ER PR links in our observations 

• We saw that total PR RNA levels were reduced on stiffer matrixes, however this may not be the whole story. Progesterone receptors (PR) are expressed as two different isoforms in breast tissue, PR-A and PR-B. PR-B is the longer isoform and PR-A is a smaller protein due to a downstream transcriptional start site in the first exon (Lanari et al., 2012). They can form homo and heterodimers and are typically expressed in equal concentrations within the normal breast. However, this expression balance is often skewed in cancer, with a higher concentration of PR-A than PR-B (Graham et al., 2005). We saw at stiffer matrixes that PR-B not PR-A was reduced. This effectively increases the ratio of A:B. We did not originally include this information because we do not currently understand it’s relevance. However it is now added in supplementary material along with expression of ER, which did not change. The role this plays will be part of future investigations. The section covering this in the discussion now reads

“Here we examined expression of PGR and it’s isoforms on different stiffnesses (Supplementary Figure 3), we saw reduction of total PGR and PGR-B in cells grown on stiffer matrixes, altering the ratio of PGR-A to PGR-B, again suggesting that the PR pathway is important in the context of stiffness. Extended studies of PR signalling in the context of stiffness will be the subject of future studies. In contrast ER expression was not altered (Supplementary Figure 3). The cell lines where resistance is observed are both ER and PR expressing. There are known links between ER and PR function especially with regards to endocrine therapy resistance (reviewed in 29). Although resistance was not seen in cell lines only expressing ER, ER expression and activation in PR/ER expressing cells was not altered on stiffer matrixes and the SPRM mifepristone abolished resistance, the study is limited by lack of inclusion of a PR only expressing cell line and we cannot exclude the possibility that cross-talk between PR and ER activity contributes to the phenotype seen.”

Reviewer #1: Authors demonstrate chemotherapy resistance in model luminal A lines, the standard of care treatment is conventionally endocrine therapy for model ER expressing cell lines. The authors should expand on and highlight the significance of sensitizing cells to chemotherapy when first line may be an endocrine therapy and not a chemotherapy.

• We agree that the translational relevance may not be immediate or obvious but the findings are novel and PR inhibitors and gemcitabine are both agents with discussed use in BC and therefore we think they are of interest to the community. This is the first study to make such links – time and further studies will reveal relevance. 

• We have added to the discussion 

 “The cell lines where resistance is observed are both ER and PR expressing. There are known links between ER and PR function especially with regards to endocrine therapy resistance (29). Although resistance was not seen in cell lines only expressing ER, ER expression in PR/ER expressing cells was not altered on stiffer matrixes and the SPRM mifepristone abolished resistance, the study is limited by lack of inclusion of a PR only expressing cell line and we cannot exclude the possibility that cross-talk between PR and ER activity contributes to the phenotype seen. Gemcitabine is not used as first line standard of care treatment in the ER/PR positive tumours that our cell lines represent. However, PR is present in 55-60% of all breast cancers (30) and clinical trials are currently underway for PR inhibition as a monotherapy (31) indicating an exciting new treatment regime. Our findings suggest that combination with gemcitabine would be an exciting translational possibility for patients bearing PR expressing tumours.”

Reviewer #2: General comments: This manuscript examined the correlation between matrix stiffness and sensitivity to chemotherapy in breast cancer using two-dimensional (2D) and three-dimensional (3D) culture systems. The authors showed that breast cancer cell growth and size increased at a higher stiffness in 2D and 3D culture systems. Interestingly, the authors found that MCF-7 and T47D breast cancer cells expressing progesterone receptors, PR, showed gemcitabine resistance at higher stiffness (4 kPa) than lower stiffness (500 Pa). Pre-treatment with a progestin antagonist, mifepristone (RU486), abolished gemcitabine when cells were grown at high stiffness (kPa). The authors suggested that selective progesterone receptor modulators (SPRM) such as mifepristone may be helpful in reducing resistance to gemcitabine in PR-positive breast tumors.

Overall, the author described an exciting discovery that PR signaling could play a role in sensitivity to a chemotherapeutic agent such as gemcitabine. However, the data presented in this manuscript is rather descriptive with little to no mechanistic detail, and some key points will need to be clarified, especially regarding PR contribution. A major revision of the manuscript is suggested.

• Reviewer 2 agrees that the data are technically sound and data support conclusions, they say that it is an exciting discovery and that the data presented in this manuscript is rather descriptive with little mechanistic detail. This PloSOne manuscript aims to present our discovery, future manuscripts will elucidate the mechanism. We address each of the specific points made below.

Specific comments

Reviewer #2 Introduction: additional information is needed. The background and significance of the study is not clear. What is known about matrix and cancer cells? What is the significance of this study?

• More context is now included to the introduction

This now includes:

“Maintaining a homeostatic environment within a tissue requires dynamic conversation between epithelial cells and the cells which reside within the surrounding interstitial matrix. This includes fibroblasts which excrete ECM components allowing ECM modulation. This fine tuning of ECM content and structure provides an important balance of tension within the tissue. In glandular tissues, such as the breast, the tensional homeostasis is compliant where the breast has an elastic modulus somewhere in the region of 100-200 Pascals (Pa) (5). In cancer however, stiffness is often heightened, where breast stiffness is increased from 100-200 Pa to ~4 kPa (5). As a consequence of altered tension, gene expression and/or cellular function can alter and Increased extracellular matrix (ECM) stiffness is associated with increased resistance to chemotherapies. Indeed in breast cancer, patients with a lower breast elastography responded better to neoadjuvant chemotherapy than those with a higher measured elastography (6-9), suggesting that stiffness may be linked to chemotherapeutic response. We hypothesise that targeting ECM stiffness or the molecular pathways altered in response to stiffness, may be provide a novel therapeutic window for specific treatment of cancer vs normal cells. Alternatively targeting stiffness induced pathways could modulate response to cytotoxic agents.”

“

Reviewer #2: How gemcitabine affects breast cancer is not known. It is not clear what the rationale for investigating the role of gemcitabine in breast cancer is.

• More context is now included to the introduction

This now includes

“The use of oestrogen receptor alpha (ERα) and epidermal growth factor receptor-2 (ERBB2/HER2) targeting agents has vastly improved outcomes for patients but for some patients and often in advanced disease chemotherapeutics such as anthracyclines and taxanes are commonly used as standard therapy. However, primary and acquired resistance to both cytotoxics occurs limiting their success. New cytotoxic treatments are now available for patients who have been previously treated with anthracyclines and taxanes including gemcitabine. Gemcitabine (Gemzar; 2′, 2′-difluorodeoxycytidine) is an analogue of deoxycytidine and a pyrimidine antimetabolite widely used in other solid tumours. In clinical trials in BC patients, gemcitabine has produced varied results perhaps linked to BC subtype (2, 3). In pancreatic cancer, where gemcitabine is a first line therapy in advanced disease, response is linked to tissue stiffness (4), but this has not been tested in BC. Given the varied response of BC to gemcitabine and to understand further the potential utility of gemcitabine in advanced BC we asked whether ECM stiffness modulates response to gemcitabine in different breast cancer cell lines.”

Reviewer #2:. Line 240. The conclusion “These results suggest that EMT is not occurring in MCF-7 cells between 500 Pa and 4 kPa.” needs to be further substantiated. The authors only examined two EMT markers in breast cancer cells and concluded that EMT was not occurring. More information is required.

• In pancreatic cancer others have seen that stiffness can promote elements of EMT, including decreases in E-cadherin expression, nuclear localisation of β-catenin, YAP and TAZ and changes in cell shape towards a mesenchymal phenotype. (https://pubmed.ncbi.nlm.nih.gov/28671675/). In breast cancer cells we did not see this – We had previously shown data of E-cadherin expression and Yap data demonstrating no change. We have now added data demonstrating b-catenin localisation is not altered. And adjusted the text to state that in contrast we did not see these aspects of EMT being altered. 

This section now reads:

“MCF-7 breast cancer cells are resistant to gemcitabine at an increased stiffness but this in not attributed to changes in EMT markers. 

Cell viability assays were used to determine the response of MCF-7 cells to gemcitabine at both 500 Pa and 4 kPa (Figure 2a). A differential response was only observed to the anti-metabolite gemcitabine, where cells cultured at 4 kPa were more resistant to gemcitabine than cells cultured at 500 Pa. Interestingly this was not observed for the other antimetabolite agents tested, 5-FU and HU (Figure 2a). In other models, stiffness has been seen to promote elements of EMT, including decreases in E-cadherin expression, nuclear localisation of β-catenin, YAP and TAZ and changes in cell shape towards a mesenchymal phenotype (10, 11). Further these changes have been linked to therapeutic response (10, 11). However, although we did see changes in cell morphology (Figure 1), we observed no difference in b-catenin localisation or E-cadherin expression between the two stiffnesses (Figure 2b&c). Similarly, there was no difference in the nuclear intensity of YAP between the two stiffnesses (Figure 2e&f). These results suggest that such EMT like characteristics are not being altered in MCF-7 cells between 500 Pa and 4 kPa. “

Reviewer #2:Additionally, EMT usually occurs in nontransformed cells. What is the significance of

• Looking at EMT in non transformed cells would be informative but beyond the scope of our initial findings. We believe that the experiments we performed are relevant because of the findings in Pancreatic cancer as discussed above. We hope that with eth revised wording above the relevance of our experiment is now clear.

Reviewer #2:. PR contribution to gemcitabine sensitivity. High dose 10-15 uM of mifepristone is needed to see the effect. Mifespristone is a PR antagonist that has a high affinity to PR and should bind to the receptor in nM range. At high concentration, mifespristone could have nonspecific effects through binding to other receptors such as glucocorticoid receptor (GR). Additionally, could protein agonist such as medroxyprogesterone or R5020 reverse mifepristone’s effect.

• The reviewer highlights important follow up experiments that will be vital to our future mechanistic understanding of the project. These are now added to the discussion.

“It should be noted that high dose 10-15 �M of mifepristone was used to see this effect. It is possible that at high concentration, mifespristone could have nonspecific effects through binding to other receptors such as glucocorticoid receptor (GR). It will be important going forward to examine other pathways by which mifepristone could be functioning. It would also be interesting to test whether protein agonists such as medroxyprogesterone or R5020 reverse mifepristone’s effect.”

---

## [Decision Letter · Decision Letter 1]

27 Apr 2022

Progesterone receptor expression contributes to gemcitabine resistance at higher ECM stiffness in breast cancer cell lines.

PONE-D-21-34392R1

Dear Dr. Bryant,

We’re pleased to inform you that your manuscript has been judged scientifically suitable for publication and will be formally accepted for publication once it meets all outstanding technical requirements.

Kind regards,

Philip C. Trackman, Ph.D.

Academic Editor

PLOS ONE

Additional Editor Comments (optional):

Reviewers' comments:

Reviewer's Responses to Questions

**Comments to the Author**

1. If the authors have adequately addressed your comments raised in a previous round of review and you feel that this manuscript is now acceptable for publication, you may indicate that here to bypass the “Comments to the Author” section, enter your conflict of interest statement in the “Confidential to Editor” section, and submit your "Accept" recommendation.

Reviewer #1: All comments have been addressed

2. Is the manuscript technically sound, and do the data support the conclusions?

Reviewer #1: Yes

3. Has the statistical analysis been performed appropriately and rigorously? 

Reviewer #1: Yes

4. Have the authors made all data underlying the findings in their manuscript fully available?

Reviewer #1: Yes

5. Is the manuscript presented in an intelligible fashion and written in standard English?

Reviewer #1: Yes

6. Review Comments to the Author

Reviewer #1: Authors have met all reviewer comments in their revision. Edits and explanation for ER expression and function have been met and no further edits are needed. Nice work on this manuscript.

7. PLOS authors have the option to publish the peer review history of their article (what does this mean?). If published, this will include your full peer review and any attached files.

Reviewer #1: No

---

## [Editor Report · Acceptance letter]

16 May 2022

PONE-D-21-34392R1 

Progesterone receptor expression contributes to gemcitabine resistance at higher ECM stiffness in breast cancer cell lines. 

Dear Dr. Bryant:

I'm pleased to inform you that your manuscript has been deemed suitable for publication in PLOS ONE. Congratulations! Your manuscript is now with our production department. 

Kind regards, 

on behalf of

Dr. Philip C. Trackman 

Academic Editor

PLOS ONE